# Antioxidant Enzyme Activities Correlated with Growth Parameters of Wheat Sprayed with Silver and Gold Nanoparticle Suspensions

**Abdul Manaf [1], Xiukang Wang [2,\*], Fatima Tariq [1], Hafiz Muhammad Jhanzab [3], Yamin Bibi [4], Ahmad Sher [5,\*], Abdul Razzaq [1], Sajid Fiaz [6], Sikander Khan Tanveer [3] and Abdul Qayyum [7,\*]**

1 Department of Agronomy, Pir Mehr Ali Shah-Arid Agriculture University Rawalpindi, Rawalpindi 46300, Pakistan; drmunaf@uaar.edu.pk (A.M.); f.tariq2001@yahoo.com (F.T.); arazzaq57@yahoo.co.in (A.R.)
2 College of Life Sciences, Yan'an University, Yan'an 716000, China
3 Wheat Program, Crop Sciences Institute, National Agricultural Research Centre, Park Road, Islamabad 44000, Pakistan; jhanzabmuhammad200@yahoo.com (H.M.J.); sikander73@hotmail.com (S.K.T.)
4 Department of Botany, Pir Mehr Ali Shah-Arid Agriculture University Rawalpindi, Rawalpindi 46300, Pakistan; dryaminbibi@uaar.edu.pk
5 College of Agriculture, Bahauddin Zakariya University, Bahadur Sub-Campus, Layyah 31200, Pakistan
6 Department of Plant Breeding, Genetics, The University of Haripur, Haripur 22620, Pakistan; sfiaz@uoh.edu.pk
7 Department of Agronomy, The University of Haripur, Haripur 22620, Pakistan
\* Correspondence: wangxiukang@yau.edu.cn (X.W.); ahmad.sher@bzu.edu.pk (A.S.); aqayyum@uoh.edu.pk (A.Q.)

**Abstract:** Application of nanotechnology is crucial for a sustainable increase in food production to cope with the increasing food demand of the burgeoning population. Wheat production has to increase significantly for food security in Pakistan with the help of nanotechnology. In biological systems, utilization of nanoparticles has been increased due to their growth-promoting effects on germination, photosynthetic attributes, nutrient use efficiency and metabolic activities. An experiment was conducted with the objective to establish a relationship between growth parameters and antioxidant enzyme activity in response to silver (Ag) and gold (Au) nanoparticles (NPs). Application of Ag (20 mg/L) and Au NPs (10 mg/L) significantly enhanced the antioxidant enzyme activities of ascorbate peroxidase, catalase and guaiacol peroxidase. Consequently, growth parameters: fresh and dry biomass, leaf area, chlorophyll (a, b) and total chlorophyll contents, also increased significantly. These results suggest that application of Ag and Au NPs has the potential to promote wheat growth through enhancing the antioxidant enzyme activities.

**Keywords:** silver and gold nanoparticles; leaf area; chlorophyll content; antioxidant enzyme

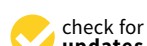



## 1. Introduction

Agriculture is facing several problems presently. Major challenges include the increasing food demand for the growing population. Consequently, the sciences and technologies which can help attain a vertical increase in crop yields are becoming more and more important. Additionally, the increased probability of physical stresses due to global warming is hindering the goal of increasing food yields [1]. Furthermore, increased use of agrochemicals is posing a serious threat to human health and the environment [2]. Therefore, producing more food, and good-quality food, in a resource-sustainable and environmentally friendly manner is the foremost challenge for agriculture in the coming decades. Target-oriented, cheaper and efficient technologies have to be employed for a sustainable increase in crop production. Innovative and knowledge-based technologies have to be explored for this purpose [3]. Nanotechnology has the potential to boost agricultural production, in order to fulfill the food, feed and fiber requirements of human beings [4].

Wheat is the main cereal crop grown in Pakistan. Wheat is grown in almost every part of the country because it is considered as a major food. It contributes about 1.7% to the gross domestic product (GDP) of Pakistan and 8.7% to the value added in agriculture [5]. Nonetheless, the yield per hectare is low in Pakistan as compared to other major wheat-producing countries. The average yield of wheat in Pakistan is 2.84 tons per hectare [5]. The wheat yield needs to be increased many folds to ensure food security in Pakistan in the near future. Better crop management will also help to enhance the wheat yield. Nanotechnology, in addition to other sciences and technologies, also has the potential to augment the wheat yield. Nanoparticles owe their importance to their special chemical reactivity, optical properties, tensile strength, electrical conductivity and magnetic properties. They have diverse applications above their bulk counterparts in several fields of science and technology such as agriculture, medicine, engineering and pharmaceuticals [6].

The application profile of nanoparticles is expanding continuously in all disciplines of life [7]. Different types of nanoparticles (NPs) have been tested in agriculture with favorable effects on germination, growth, physiological activities, water and fertilizer use efficiency [8–10]. They have also been found to enhance root growth branching, biomass and photosynthetic pigments [9,11]. Enhanced nutrient use efficiency and photosynthetic activity ultimately result in an increased yield and a better grain quality [12,13]. Nanoparticles have also been found to have antioxidant activity [14]. Extensive experimentation has been carried out to study the effects of metal NPs such as silver, copper, zinc, gold and titanium [14–16] on crop plants. Significant changes in plants' morphology were observed after exposure to metal NPs. Determination of the antioxidant capacity of metal nanoparticles is very crucial. The aim of this study was to analyze the protective effect of Ag and Au NPs against oxidative stress in relation to biomass, leaf area and the chlorophyll content of wheat seedlings.

## 2. Materials and Methods

The lab experiment was carried out in the Crop Physiology Laboratory, Department of Agronomy, PMAS-Arid Agriculture University Rawalpindi, Pakistan, to compare the antioxidant activity of Ag and Au NPs correlated with three enzymatic activities, namely, ascorbate peroxidase, catalase and guaiacol peroxidase.

### 2.1. Synthesis of Silver Nanoparticles

Silver nanoparticles were synthesized chemically by reducing silver nitrate ($AgNO_3$) with tri-sodium citrate dihydrate ($HOC(COONa)(CH_2COONa)_2 \cdot 2H_2O$) [17]. A stock solution of 50 mg/L of silver nanoparticles was prepared by reducing 80 mg of $AgNO_3$ solution with a solution containing 40 mg of tri-sodium citrate. Silver nitrate solution was heated and stirred at 5000 rpm on a magnetic stirrer at 80 °C. When the temperature reached 80 °C, tri-sodium citrate solution was added to silver nitrate solution with continuous stirring and heating until the color of the solution turned to golden yellow. The golden yellow color is an indication of the formation of Ag NPs. Further dilutions were prepared from this stock solution. The size of Ag NPs was analyzed by scanning electron microscopy (JOEL-JSM-6490LA™ SEM) at the University of Peshawar, Pakistan. The size of Ag NPs ranged from 20 to 30 nm (Figure 1).

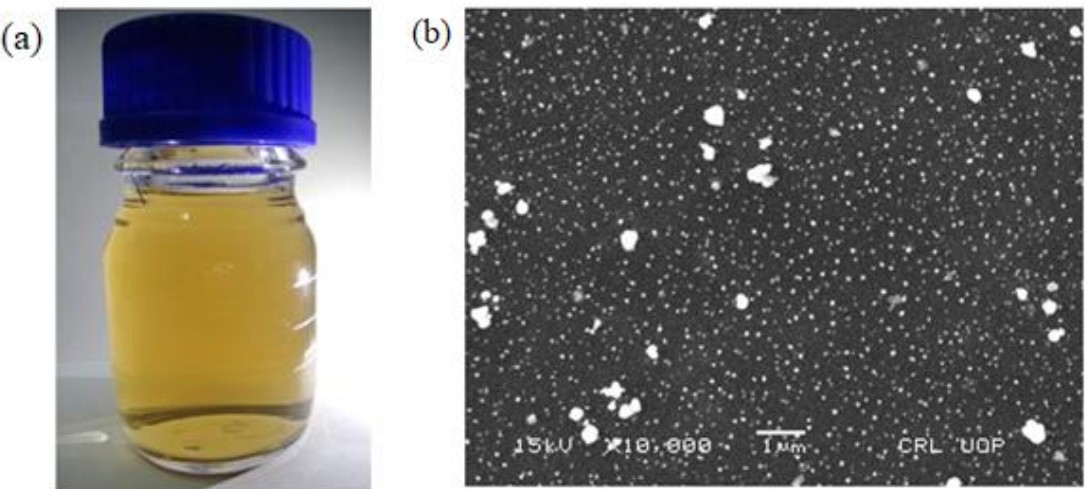

**Figure 1.** Synthesis and characterization of Ag NPs: (**a**) 20–30 nm silver nanoparticles; (**b**) SEM of 20–30 nm silver nanoparticles. Representative SEM, batch-to-batch variation may be expected.

### 2.2. Synthesis of Gold Nanoparticles

Gold nanoparticles were synthesized according to the method described by Zhao et al. [18], with slight modifications. A stock solution of 40 mg/L of Au NPs was prepared by reducing 40 mg of chloro-auric acid solution with a solution containing 41 mg of tri-sodium citrate. Chloro-auric acid solution was heated in a flask (kept in water) and stirred at 5000 rpm on a magnetic stirrer at 80 °C. When the temperature reached 80 °C, tri-sodium citrate solution was added to chloro-auric acid solution with continuous stirring and heating until the color of the solution turned red. The red color of the solution is an indication of the formation of Au NPs. Gold nanoparticles were analyzed by SEM (JOEL-JSM-6490LA™ SEM) at the University of Peshawar, Pakistan. Working solutions were prepared from this stock solution for experimental use. The size of Au NPs ranged from 5 to 20 nm (Figure 2).

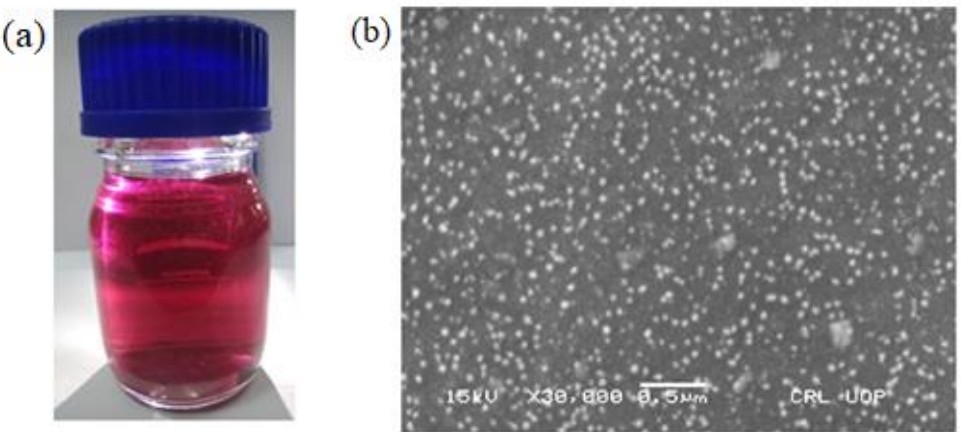

**Figure 2.** Synthesis and characterization of gold nanoparticles: (**a**) 5–20 nm gold nanoparticles; (**b**) SEM of 5–20 nm gold nanoparticles. Representative SEM, batch-to-batch variation may be expected.

### 2.3. Seed Germination

Seeds of the wheat variety 'Punjab-2011' were collected from the Wheat Programme, Crop Sciences Institute, National Agricultural Research Centre (NARC), Islamabad, Pakistan. Healthy seeds were selected and dipped into hydrogen peroxide ($H_2O_2$) solution for 15 min, followed by rinsing with distilled water for 4 times. The seeds were then placed in Petri dishes fitted with 4 layers of moistened filter paper for germination.

*2.4. Transplanting of Seedlings*

About 1 L solution of Murashige and Skoog (MS) medium was filled in plastic pots, and the pH of the solution was set at 5.8 [19]. The thermo-pole sheets were cut to fit in the pots, and holes were made in the sheet for the transfer of plants.

Seedlings were allowed to grow up to 7–10 cm in length. The seedlings were then fixed into the holes with the help of a thin foam sheet. Air pumps were used for continuous aeration of the solution in pots. The pots were put under light of 6000–7000 lux and a temperature of about 26–28 °C. The solutions were changed after 7 days. Transplanted seedlings were allowed to grow for 10 days. Different concentrations of Ag (10, 20, 30 mg/L) and Au NPs (10, 20, 30 mg/L) were used for the foliar spray of the treatments. Only the spray of distilled water served as a control treatment. A completely randomized design (CRD) with four replications was employed for the layout of the experiment and statistical analysis.

After 2 weeks of spray with NPs, data for leaf area [20], chlorophyll content, fresh weight, dry weight and antioxidant activity of enzymes (ascorbate peroxidase, catalase and guaiacol peroxidase) were recorded. Chlorophyll analysis was determined by following the method of Arnon [21]. Ethanol (80%) at 5 mL was taken in test tubes and immediately weighed (0.1 g), fresh leaf samples were added, immersed in ethanol, and tubes were capped. The extract was kept in a water bath at 80 °C for 10 min. The extract was cooled in a dark room. The optical density was measured at 663 and 645 nm for chlorophyll 'a' and 'b', respectively, by exposure to lower light by using a UV spectrophotometer. Ethanol (80%) was used as a blank sample.

$$\text{Chlorophyll a} = \frac{(12.7 \times \text{OD at } 663 - 2.69 \times \text{OD at } 645) \times V}{1000 \times \text{Fresh Shoot Weight (g)}} \tag{1}$$

$$\text{Chlorophyll b} = \frac{(22.9 \times \text{OD at } 645 - 4.69 \times \text{OD at } 663) \times V}{1000 \times \text{Fresh Shoot Weight (g)}} \tag{2}$$

$$\text{Total Chlorophyll} = \frac{(20.2 \times \text{OD at } 645 + 8.02 \times \text{OD at } 663) \times V}{1000 \times \text{Fresh Shoot Weight (g)}} \tag{3}$$

where V = volume of extract (mL), and OD = optical density.

Ascorbate peroxidase activity was measured by monitoring the decline in absorbance at 290 nm, as ascorbate ($\varepsilon = 2.8$ mM$^{-1}$ cm$^{-1}$) was oxidized for 3 min [22]. Catalase activity was determined by measuring the decreasing rate in the absorbance of $H_2O_2$ at 240 nm [23]. Guaiacol peroxidase activity was measured using a reaction medium containing 50 mM phosphate buffer (pH 7), 9 mM guaiacol and 19 mM $H_2O_2$ [24].

*2.5. Experimental Design and Statistical Analysis*

Data obtained were analyzed for analysis of variance (ANOVA) by a two-factor factorial in a completely randomized design, and mean comparisons ($p \leq 0.05$) were carried out by the Duncan multiple range (DMR) test using M-Stat-C Statistical software [25].

## 3. Results

*3.1. Characterization of Nanoparticles*

Silver and gold nanoparticles were chemically synthesized. Synthesized Ag and Au NPs were characterized through SEM for their size measurement. The results reveal that Ag NPs are uniform in shape, and their size ranges from 20 to 30 nm (Figure 1), whereas Au NPs' size ranges from 5 to 20 nm (Figure 2).

*3.2. Growth of Wheat Seedlings in Response to Silver and Gold Nanoparticles*

The growth of wheat seedlings exposed to Ag NPs was significantly better than that of the control seedlings. The data regarding the effect of Ag and Au NPs on the fresh weight of wheat plants are presented in Table 1. The analysis of variance exhibited highly significant differences in the fresh weight of plants among control and Ag/Au NP-treated

plants. The highest average fresh weight was recorded in plants sprayed with 20 mg/L of Ag NPs, followed by 30 mg/L and 10 mg/L of Ag NPs, as compared with control plants. However, plants treated with 10 mg/L of Au NPs also produced a significantly higher fresh weight in comparison to control plants but significantly less than the plants treated with Ag NPs. However, when the plants were sprayed with 20 mg/L and 30 mg/L of Au NPs, the lowest fresh weight was recorded, which was also obviously less than that of the control plants.

**Table 1.** Leaf fresh weight, leaf dry weight and leaf area of wheat seedlings in response to silver and gold nanoparticles.

| Treatments | Fresh Weight of Leaf (g) | Dry Weight of Leaf (g) | Leaf Area ($cm^2$) |
|---|---|---|---|
| Control | 1.700 ± 0.01 [e] | 0.397 ± 0.009 [e] | 8.250 ± 0.15 [d] |
| 10 mg/L Ag NPs | 2.927 ± 0.02 [c] | 0.623 ± 0.005 [c] | 10.613 ± 0.13 [bc] |
| 20 mg/L Ag NPs | 4.717 ± 0.05 [a] | 0.890 ± 0.001 [a] | 12.690 ± 0.10 [a] |
| 30 mg/L Ag NPs | 3.620 ± 0.04 [b] | 0.710 ± 0.004 [b] | 11.307 ± 0.11 [b] |
| 10 mg/L Au NPs | 2.027 ± 0.02 [d] | 0.543 ± 0.006 [d] | 10.047 ± 0.13 [c] |
| 20 mg/L Au NPs | 1.233 ± 0.01 [f] | 0.147 ± 0.009 [f] | 6.707 ± 0.17 [e] |
| 30 mg/L Au NPs | 0.877 ± 0.01 [g] | 0.097 ± 0.009 [f] | 5.300 ± 0.19 [f] |
| LSD Value ($p \leq 0.05$) | 0.285 | 0.065 | 1.000 |

The values represent the averages (±SE) of four independent replicates, and those followed by different letters within columns are significantly different at $p \leq 0.05$ according to Duncan's multiple range test.

The data regarding the effect of Ag and Au NPs on the dry weight of wheat plants are presented in Table 1. The analysis of variance showed highly significant differences in the dry weight of plants among control and Ag/Au NP-treated plants. The highest average dry weight was recorded in plants sprayed with the 20 mg/L concentration of Ag NPs, followed by 30 mg/L and 10 mg/L, as compared with control plants. On the other hand, plants treated with 10 mg/L of Au NPs also produced a significantly higher dry weight in comparison to the control plants but significantly less than the plants treated with Ag NPs. However, the plants that were sprayed with 20 mg/L and 30 mg/L of Au NPs reflected a non-significant relationship with each other.

The data pertaining to the leaf area affected by Ag and Au NPs applied to wheat (Table 1) illustrate a highly significant difference among different treatments. However, the plants treated with the 20 mg/L concentration of Ag NPs produced a significantly higher leaf area as compared with the control plants. Nonetheless, the further increase in the concentration of Ag NPs showed a significant decline in leaf area but still higher than the controls. When the plants were treated with 10 mg/L of Au NPs, the leaf area of wheat was significantly higher in comparison to the controls. The further increase in the concentration of Au NPs caused a significant reduction in leaf area. The leaf area produced in response to 20 mg/L and 30 mg/L of Au NPs was significantly less than that of other plants.

The effect of Ag and Au NPs on the wheat chlorophyll content is depicted in Table 2. The highest chlorophyll 'a' content was recorded when plants were treated with 20 mg/L of Ag NPs, in comparison to control plants, followed by 30 mg/L and 10 mg/L of Ag NPs. Nonetheless, 10 mg/L of Au NPs had a significant effect in comparison with the controls. The effect of 20 mg/L and 30 mg/L of Au NPs was negative on the chlorophyll 'a' content of wheat seedlings. A significantly lower content of chlorophyll 'a', as compared to the controls, was accumulated in response to 20 mg/L and 30 mg/L of Au NPs. This indicates the phytotoxic effect of Au NPs. The accumulation of the chlorophyll 'b' content was enhanced due to the application of Ag NPs and decreased significantly due to the Au NPs, as compared to the control plants. The highest chlorophyll 'b' content was recorded when wheat seedlings were sprayed with 20 mg/L of Ag NPs. No significant differences were found in the chlorophyll 'b' content in response to 10 mg/L and 30 mg/L of Ag NPs. However, the chlorophyll 'b' content also increased significantly in these two treatments compared to the controls. Gold nanoparticles at 10 mg/L did not increase the chlorophyll 'b' content, as compared to the control, whereas the application of 20 mg/L and 30 mg/L of

Au NPs resulted in a significant reduction in the chlorophyll 'b' content compared to that of the controls. The highest total chlorophyll contents were recorded with the application of 20 mg/L of Ag NPs. The further increase in the level of Ag NPs, however, caused a significant increase in the total chlorophyll content as compared to control, but the effect was significantly less than that of 20 mg/L of Ag NPs. Higher total chlorophyll contents were recorded by the application of 10 mg/L of Au NPs compared to those of the controls. When the wheat seedlings were subjected to higher concentrations of Au NPs (20 mg/L and 30 mg/L), significantly lower total chlorophyll contents were recorded, as compared to the controls.

**Table 2.** Chlorophyll contents of wheat seedlings in response to silver and gold nanoparticles.

| Treatments | Chlorophyll 'a' ($\mu$g g$^{-1}$ Fresh Weight) | Chlorophyll 'b' ($\mu$g g$^{-1}$ Fresh Weight) | Total Chlorophyll ($\mu$g g$^{-1}$ Fresh Weight) |
|---|---|---|---|
| Control | 27.560 ± 0.48 [e] | 26.450 ± 0.50 [cd] | 54.010 ± 0.66 [e] |
| 10 mg/L Ag NPs | 30.297 ± 0.41 [c] | 30.280 ± 0.55 [b] | 60.577 ± 0.43 [c] |
| 20 mg/L Ag NPs | 33.272 ± 0.45 [a] | 33.977 ± 0.41 [a] | 67.249 ± 0.61 [a] |
| 30 mg/L Ag NPs | 32.123 ± 0.50 [b] | 31.583 ± 0.46 [b] | 63.706 ± 0.55 [b] |
| 10 mg/L Au NPs | 29.137 ± 0.41 [d] | 28.043 ± 0.49 [c] | 57.180 ± 0.59 [d] |
| 20 mg/L Au NPs | 26.053 ± 0.44 [f] | 24.610 ± 0.53 [d] | 50.663 ± 0.60 [f] |
| 30 mg/L Au NPs | 24.917 ± 0.49 [g] | 20.293 ± 0.43 [e] | 45.210 ± 0.47 [g] |
| LSD Value ($p \leq 0.05$) | 0.907 | 1.863 | 3.051 |

The values represent the averages (±SE) of four independent replicates, and those followed by different letters within columns are significantly different at $p \leq 0.05$ according to Duncan's multiple range test.

### 3.3. Antioxidant Enzyme Activity as Influenced by Silver and Gold Nanoparticles

Data regarding the effect of Ag and Au NPs on the antioxidant enzyme activity of wheat plants are presented in Table 3. The data indicate highly significant ($p \leq 0.05$) differences among the antioxidant enzyme activity in response to Ag and Au NP application. The highest number of enzyme units of ascorbate peroxidase was recorded when wheat seedlings were sprayed with 20 mg/L of Ag NPs, followed by 30 mg/L of Ag NPs and 20 mg/L of Au NPs. The number of enzyme units induced by 20 mg/L of Ag NPs, 30 mg/L of Ag NPs and 20 mg/L of Au NPs was significantly higher compared to the controls. Treatment of wheat seedlings with 10 mg/L of Au NPs could not significantly increase the enzyme units of ascorbate peroxidase, as compared to the controls.

**Table 3.** Effect of silver and gold nanoparticles on antioxidant enzyme activity (enzyme units) of wheat seedlings.

| Treatments | Ascorbate Peroxidase * | Catalase ** | Guaiacol Peroxidase *** |
|---|---|---|---|
| Control | 192.00 ± 1.69 [e] | 290.00 ± 1.47 [d] | 518.33 ± 2.01 [d] |
| 10 mg/L Ag NPs | 219.67 ± 1.58 [c] | 575.00 ± 1.77 [c] | 600.00 ± 1.88 [c] |
| 20 mg/L Ag NPs | 352.33 ± 1.60 [a] | 936.00 ± 1.64 [a] | 745.67 ± 2.31 [a] |
| 30 mg/L Ag NPs | 244.67 ± 1.67 [b] | 597.67 ± 1.51 [b] | 671.67 ± 1.91 [b] |
| 10 mg/L Au NPs | 199.67 ± 1.56 [de] | 565.67 ± 1.80 [c] | 485.67 ± 1.78 [e] |
| 20 mg/L Au NPs | 239.33 ± 1.64 [b] | 165.67 ± 1.61 [f] | 354.67 ± 1.95 [f] |
| 30 mg/L Au NPs | 208.33 ± 1.51 [cd] | 259.00 ± 0.47 [e] | 293.33 ± 1.99 [g] |
| LSD Value ($p \leq 0.05$) | 15.137 | 14.652 | 10.673 |

* One unit of enzyme activity is defined as a decrease in absorbance of 0.001/min at 290 nm. ** One unit of enzyme activity is defined as a decrease in absorbance of 0.001/min at 240 nm. *** One unit of enzyme activity is defined as an increase in absorbance of 0.001/min at 510 nm. The values represent the averages (±SE) of four independent replicates, and those followed by different letters within columns are significantly different at $p \leq 0.05$ according to Duncan's multiple range test.

The highest number of enzyme units of catalase was found at 20 mg/L of Ag NPs, followed by 30 mg/L of Ag NPs and 10 mg/L of Ag NPs, in that order. The numbers of enzyme units in response to these levels of Ag NPs were significantly higher than those of the controls. Gold nanoparticles at 10 mg/L also increased the catalase enzyme activity

significantly over the controls. However, the enzyme activity at this concentration of Au NPs was significantly less than the activity induced by 30 mg/L of Ag NPs. The further increase in the concentration of Au NPs was accompanied by a significant reduction in enzyme activity compared to that of the control plants. Guaiacol peroxidase enzyme activity in wheat leaves was the highest with the application of 20 mg/L of Ag NPs, followed by 30 mg/L and 10 mg/L, as compared to the activity of the enzyme in control plants. Nonetheless, the activity of guaiacol peroxidase decreased significantly when the plants were sprayed with 10 mg/L, 20 mg/L and 30 mg/L of Au NPs, indicating a phytotoxic impact of Au NPs at these levels.

The correlations between the growth parameters and antioxidant enzyme activity are presented in Table 4. A significant relationship was observed between all growth parameters and the antioxidant enzyme activity. Ascorbate peroxidase indicated a positive correlation with all growth parameters, whereas catalase and guaiacol peroxidase indicated a strong positive correlation with all growth parameters. Therefore, the strong positive correlation between growth and antioxidant enzyme activities may lead to an increase in the growth and development of the plants.

**Table 4.** Correlation between growth parameters and antioxidant enzyme activity.

|  | Ascorbate Peroxidase | Catalase | Guaiacol Peroxidase |
|---|---|---|---|
| Fresh weight of leaf | 0.752 * | 0.913 ** | 0.974 ** |
| Dry weight of leaf | 0.609 * | 0.945 ** | 0.975 ** |
| Leaf Area | 0.593 * | 0.924 ** | 0.963 ** |
| Chlorophyll a | 0.657 * | 0.927 ** | 0.976 ** |
| Chlorophyll b | 0.635 * | 0.879 ** | 0.973 ** |
| Chlorophyll total | 0.749 * | 0.932 ** | 0.973 ** |

* Correlation is significant at $\alpha$ 0.05. ** Correlation is highly significant at $\alpha$ 0.01.

## 4. Discussions

An efficient approach to the production of fine NPs with the same physiochemical properties is through chemical reduction. Chemically synthesized Ag NPs were characterized through X-ray diffraction analysis, and their size ranged from 10 to 20 nm [8]. Jhanzab et al. [26] reported that Ag NPs were synthesized through chemical reduction, and their size was analyzed through X-ray diffraction, returning a value of < 20 nm. Many procedures have been established for the synthesis of NPs, e.g., chemical, physical and biological. In a chemical reduction procedure, the solution of the metal is reduced in a way so that no metal ions are present in the system [27]. The chemical reduction process produced a relatively high yield, a low cost and an easy practice for the synthesis of large-scale NPs [28].

Razzaq et al. [8] reported that a sufficient increase in the concentrations of Ag and Au NPs, i.e., 10, 20 and 30 mg/L, results in a higher fresh weight and dry weight than in the controls. However, 20 mg/L of Ag NPs potentially increased the fresh and dry weight. If the concentrations of Ag NPs and Au NPs are further increased to 30 mg/L and 50 mg/L, a gradual reduction in the fresh and dry weight of wheat is observed. An increase in the concentration and prolonged exposure caused a reduction in the leaf area and fresh and dry weight. The plants treated with 25 mg/L of Ag NPs produced a remarkably higher root biomass than the controls or those treated with other levels of Ag NPs. Favorable effects on the fresh weight, dry weight and root biomass of wheat seedlings were observed when wheat plants were treated with 20 mg/L of Ag NPs in MS medium [8]. However, there was a decrease in the fresh and dry weight when the plants were exposed to higher concentrations of Ag NPs. In our experiment, the highest increase in the fresh weight of plants was recorded when the plants were exposed to 20 mg/L of Ag NPs and 10 mg/L of Au NPs. Therefore, the results of this experiment show that the application of Ag NPs at 20 mg/L has the maximum potential to enhance the fresh weight of the plants, followed by 10 mg/L of Au NPs.

Nanoparticle concentration-dependent effects have been reported in several studies; however, nanoparticles enhance growth at lower doses [29]. Other NPs also have analogous properties. Similar results of Ag NPs were also observed in the experiment. Elevated concentrations had negative effects, whereas lower concentrations boosted the growth of wheat crops. The reactant properties of Ag NPs are extraordinary and higher than silver nitrate [30]. Silver nanoparticles restrain the activity of ethylene in plants by restricting silver on the locales which are typically bound by ethylene [31]. Silver nanoparticles can empower plants to repress senescence caused by responsive oxygen species' (ROS) age because of oxidative stress. An increase in the concentration and prolonged exposure caused a reduction in the leaf area, fresh weight and dry weight. Favorable effects on the fresh weight, dry weight and root biomass of wheat seedlings were observed when wheat plants were treated with 20 mg/L of Ag NPs in MS medium [8]. However, there was a decrease in the fresh and dry weight when the plants were exposed to higher concentrations of Ag NPs. In our experiments, the highest increase in the dry weight of plants was recorded when the plants were exposed to 20 mg/L of Ag NPs and 10 mg/L of Au NPs. Therefore, the results of this experiment clearly indicate that the application of Ag NPs at 20 mg/L has the maximum potential to enhance the dry weight of the plants, followed by 10 mg/L of Au NPs.

Xiumei et al. [32] reported that the peanut leaf area increased by the use of nanocalcium carbonate. A comparative reaction for the increment in the leaf area of wheat plants was seen by the utilization of 20 mg/L of Ag NPs in our study. A significant increase in the growth of tomato, spinach and peanut plants was observed in response to application of carbon nanotubes, iron oxide NPs, nanotitanium and nanocalcium carbonate [32–34]. In our experiment, the highest increase in leaf area was recorded when plants were treated with 20 mg/L of Ag NPs. Silver nanoparticles seem to have a promising effect on the chlorophyll content, whereas Au NPs have a negative effect on the chlorophyll content. Silver nanoparticles at a 20 mg/L concentration induced the highest accumulation of chlorophyll content, which was significantly higher than that in control seedlings. However, Au NPs had adverse effects on the chlorophyll content of wheat. Consequently, Ag NPs at 20 mg/L may be used to enhance the accumulation of chlorophyll content and, in turn, the photosynthesis and yield of the crop. Razzaq et al. [8] recorded significantly higher total chlorophyll as well as chlorophyll 'a' and 'b' contents in response to 10, 20 and 30 mg/L of Ag NPs when compared with the controls. However, the effect of 20 mg/L of Ag NPs was more prominent and produced the highest chlorophyll a and b and total chlorophyll contents. An increase in the Ag NP concentration caused a decline in the chlorophyll contents of the wheat seedlings. Variable responses of several plants to Ag NPs have been accounted for by different scientists. Silver nanoparticles initiate noteworthy changes at the molecular and physiological level in this manner, influencing plant growth. Applying Ag NPs to soil significantly influences the growth of wheat. The use of 50 mg/L of Ag NPs expanded the plant stature, fresh and dry weight and the length of roots and shoots of wheat seedlings [35] and the fresh weight, root and shoot length, vigor index and chlorophyll substance of seedlings of *Brassica juncea* [36]. Silver nanoparticles can cause faster development (0.5–1000 mg/L) and can possibly fundamentally increment the plant stature (25–50 mg/L) and fresh and dry weight over the controls in wheat [36]. Silver nanoparticles at 50 mg/L were seen to cause amazing increments in the total chlorophyll content, chlorophyll a, chlorophyll b and root fresh weight in wheat plants, yet they did not influence the shoot fresh weight [35]. Utilization of 50 mg/L of nanosilver in a blend with nitrogen and nitroxin expanded the weight and yield of potato tubers. We discovered a noteworthy increment in the fresh weight, dry weight and chlorophyll contents of wheat with 25 mg/L of soil-connected Ag NPs. These ideal impacts of soil-connected Ag NPs on development might be because of the lower bioavailability and amassing in plants along these lines, fortifying development.

A few examinations revealed the negative impacts of Ag and Au NPs. Critical diminishments in root extension, chlorophyll substance and shoot and root fresh weight were

noted on the introduction of rice seedlings to Au NPs. The chlorophyll content, shoot dry weight and root dry weight of rice were fundamentally inhibited at 1000 mg/L of Au NPs [37]. Retained Au NPs upset the thylakoid film structure, diminished the chlorophyll content and inhibited the root length and fresh weight [38]. In any case, antagonistic impacts are connected with higher fixation and greater bioavailability. Every one of the investigations revealing negative impacts led analyses in arrangement or utilized higher fixation. Authors who utilized Ag NPs in the soil at low concentrations [35,36] discovered beneficial outcomes of Ag NPs. Silver nanoparticles had no extreme harmful impacts and improved the peroxidase and catalase activity [39], although Ag NPs applied to duckweed (*Spirodela polyrhiza*) in a nutrient mixture essentially diminished the plant biomass and chlorophyll content [40]. The impact of Ag NPs appears to be concentration-dependent. Nanoparticles accelerate growth at low measurements, yet they hinder development at a high dosage [29]. Different NPs likewise have comparative impacts. Iron-based NPs at a lower dosage advanced the growth of maize; however, they hindered growth at higher concentrations [41]. We additionally assessed comparable outcome impacts of Ag NPs in our investigation. The higher dosage of Ag NPs had inhibitory impacts, while the lower dosage upgraded the development of wheat. Salama [42] observed that expanding the concentration of Ag NPs from 20 to 60 mg/L prompted an expansion in the shoot and root lengths, leaf surface area and chlorophyll, starch and protein substance of common bean and corn. An extra increment in the level of Ag NPs brought about a decrease in these parameters. Soil use of Ag NPs at low concentrations can upgrade the development of plants, while Ag NPs connected through nutrient solution increment the bioavailability and aggregation in plants, consequently hindering development.

A significant increase in the activity of the ascorbate peroxidase, catalase and guaiacol peroxidase enzymes was observed at low concentrations of Ag (20 mg/L) and Au (10 mg/L) NPs. However, with the increased concentration of both metal NPs, the activities of these enzymes were markedly reduced. The activity of antioxidative enzymes may be a good marker of the toxic influence of the external environment on plants. Mehrian et al. [43] studied the influence of Ag NPs on the activity of antioxidant enzymes in *Lycopersicon esculentum*. It was found that, with the increase in the concentration of Ag NPs in the plant tissue medium, the activity of catalase and peroxidase in the shoots and roots increased. The increased activity of catalase and peroxidase was also observed in the callus tissue of sugar cane propagated on media with the addition of 20 to 60 mg/L of Ag NPs [44]. Iqbal et al. [45] carried out a study to investigate the effect of Ag NPs on the physiological, biochemical and antioxidant parameters of wheat (*Triticum aestivum* L.) under heat stress conditions. It was found that the addition of Ag NPs had a protective effect on plant tissues under stress conditions. Wheat plants treated with Ag NPs showed a significant increase in dry matter, with a simultaneous increase in superoxide dismutase (SOD), peroxidase (POX), catalase (CAT) and ascorbate peroxidase (APX) activity under heat stress conditions. In this study, we analyzed the impact of the addition of Au and Ag NPs on the activity of the CAT, APX and GP (guaiacol peroxidase) antioxidative enzymes. The conducted study revealed a significant impact of the applied nanoparticles on the activity of all the tested antioxidative enzymes.

Significant modifications in plants' morphology were observed after exposure to Ag and Au NPs. Plant growth, germination, leaf area and biomass are commonly used morphological attributes for assessing the growth-promoting activity of Ag NPs [46–48]. At the physiological level, the activity of Ag NPs in plants is anticipated by increasing the chlorophyll content, resulting in enhanced photosynthesis and thus promoting plant growth and hormonal activity [48]. At higher concentrations, Ag NPs can also cause toxicity at the cellular and molecular level in plants; however, various studies indicated that NPs, especially Ag NPs, have a promoting effect on plant growth and physiology [49]. Nanometal particles show a strong affinity to plant tissues, and they activate enzymatic pathways responsible for the production of secondary metabolites [50]. They also contribute to the peroxidation of cellular membranes in plant cells and affect the expression of genes

responsible for the production of biologically active compounds [50]. The main mechanism underlying the phyto-stimulatory effect of Ag NPs is the production of excess reactive oxygen species (ROS) induced by Ag NPs, resulting in oxidative stress in plant cells [51–53]. Silver nanoparticles promoted plant growth and development through regulation of energy metabolism by suppression of glycolysis-associated proteins [54], and proteins related to secondary metabolism, protein synthesis and transport were increased [49].

**5. Conclusions**

Silver and gold nanoparticles were synthesized chemically. The results reveal that application of silver nanoparticles (20 mg/L) significantly enhanced the growth and antioxidant enzymatic activity, whereas gold nanoparticles showed an adverse effect on the growth and enzyme activity of wheat seedlings. Ultimately, it can be concluded that silver nanoparticles have significant potential to enhance antioxidant enzyme activity. This significant increase in the activity of antioxidant enzymes might be the underlying mechanism for the improved growth of wheat in response to silver nanoparticles.

**Author Contributions:** A.R. conceived the idea. A.M., F.T., H.M.J. and Y.B. conducted the experiment and collected the literature review. A.Q., A.S., S.F. and S.K.T. provided technical expertise to strengthen the basic idea. S.F., X.W. and A.Q. helped in statistical analysis. A.R., S.K.T. and A.Q. proofread and provided intellectual guidance. All authors read the first draft, helped in revision and approved the article.

**Funding:** The publication of the present work is supported by the Natural Science Basic Research Program of Shaanxi Province (grant no. 2018JQ5218) and the National Natural Science Foundation of China (51809224), Top Young Talents of Shaanxi Special Support Program.

**Institutional Review Board Statement:** Not Applicable.

**Informed Consent Statement:** Not Applicable.

**Data Availability Statement:** Not Applicable.

**Conflicts of Interest:** The authors declare no conflict of interest.

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
