# Peer review of "Antioxidant Enzyme Activities Correlated with Growth Parameters of Wheat Sprayed with Silver and Gold Nanoparticle Suspensions"

_agronomy, doi:10.3390/agronomy11081494_

Round 1
Reviewer 1 Report
- Interesting work, although in this area there are a lot of research concerning the influence of Nanoparticles on the growth, dry and fresh weight of plants. Value added is the effect of silver and gold Nanoparticles on the activity of antioxidant enzymes.
Comments:
Page 1: explain / correct the term: mushrooming population? Better will be growing population. - Results and discussion: author often uses the term Nanoparticles of silver and gold. In my opinion they can be abbreviated as AgNPs or AuNPs, which will be easier to read.
Other comments and corrections made in the text of the manuscript - Literature must be corrected in accordance with the journal guidelines as there are many errors.

Author Response
Dated: 17th July 2021
Dear Editor,
Greetings,
Thank you very much for your time and comments regarding our manuscript (agronomy-1272414). Our manuscript “Relationship of Antioxidant Enzyme Activity with Growth Parameters in Response to Silver and Gold Nanoparticles in Wheat (Triticum aestivum L.)” has been revised carefully and here we are giving our response to the reviewers’ comments. We have improved the manuscript according to the reviewers’ comments and suggestions. All the revisions can be easily identified from manuscript highlighted with BLUE color.
Once again thanks for your co-operation and valuable comments and suggestion. Moreover, the efforts of the reviewer are highly appreciated. We are hoping for pleasant response and further good comments (if any) from your side.
Dr. Abdul Qayyum
Department of Agronomy
The University of Haripur 22620 Pakistan
***************************************************************************
We are thankful to editor and reviewers for timely completion of review process and providing us with valuable feedback.
Reviewer 1
According to reviewer guidelines all the corrections have been made accordingly. Detail of point to point indications and corrections have been given below.
Comments: Page 1: explain / correct the term: mushrooming population? Better will be growing population.
Reply: Incorporated as suggested and highlighted with red color. See line # 43.
Comment: Results and discussion: author often uses the term Nanoparticles of silver and gold. In my opinion they can be abbreviated as AgNPs or AuNPs, which will be easier to read.
Reply: Abbreviations used throughout the text now and highlighted.
Comments: Other comments and corrections made in the text of the manuscript.
Reply: All suggestions are incorporated as suggested by the reviewer.
Comment: Literature must be corrected in accordance with the journal guidelines as there are many errors.
Reply: Literature is according to journal format.
Comments Suggested in Main File
Comment 1
Mushrooming Population has been replaced and corrected. Highlighted with red color. See line # 43.
Comment 2
Word “Vertically” removed. See line # 63.
Comment 3
Suggested as proposed. See line # 88-89.
Comment 4
Suggested as proposed. See line # 91.
Comment 5
Reference corrected according to journal format. See line # 95.
Comment 6
“Punjab -2011” is corrected. See line # 122.
Comment 7
Good removed. Now the sentence is Healthy seeds…… See line # 124.
Comment 8
Petri Corrected. See line # 127.
Comment 9
MS (Murashige and Skoog Medium) corrected. See line # 130.
Comment 10
Corrected antioxidant activity of enzymes. See line # 146-147.
Comment 11
Jhanzab et al. (2015) corrected. See line # 183.
Comment 12
Word “over” replaced with “than in”. See line # 203.
Comment 13
Words “Weight” and “plant” removed. See line # 206-207.
Comment 14
Word “from” replaced with “than in” removed. See line # 209.
Comment 15
Words “vice versa” removed and sentence is modified. See line # 230-232.
Comment 16
Words “wheat plants” modified to wheat. See line # 253.
Comment 17
Words “control plants” modified to control. See line # 260.
Comment 18
Words “were induced” modified to “had significant”. See line # 275.
Comment 19
Word “less” is replaced with “less content”. See line # 277.
Comment 20
Words “was at par with one another” were replaced and sentence is modified. See lines # 284-285.
Comment 21
Word “over” is replaced with “than in”. See line # 286.
Comment 22
Sentence “Total chlorophyll content accumulated in response to application of 10 ppm gold nanoparticles was more to that of control plants” is restructured and modified. See line # 294-295.
Comment 23
Word “that of” is replaced with “in”. See line # 303.
Comment 24
Words “and” is replaced with “as well as” and sentence is modified. See lines # 306-309.
Comment 25
Botanical name Italic. See lines # 320.
Comment 26
Words “speedier” modified to “faster”. See line # 321.
Comment 27
Words “advance” modified to “accelerate”. See line # 349.
Comment 28
Words “pertaining to” modified to “showing the”. See line # 365.
Comment 29
Words “wheat seedlings not sprayed with any nanoparticle” modified to “control”. See line # 373.
Comment 30
Units provided. See line # 403.
Comment 31
All the references have been rechecked and corrected.
Reviewer 2 Report
Please see the attachment.

Author Response
Dated: 17th July 2021
Dear Editor,
Greetings,
Thank you very much for your time and comments regarding our manuscript (agronomy-1272414). Our manuscript “Relationship of Antioxidant Enzyme Activity with Growth Parameters in Response to Silver and Gold Nanoparticles in Wheat (Triticum aestivum L.)” has been revised carefully and here we are giving our response to the reviewers’ comments. We have improved the manuscript according to the reviewers’ comments and suggestions. All the revisions can be easily identified from manuscript highlighted with RED color.
Once again thanks for your co-operation and valuable comments and suggestion. Moreover, the efforts of the reviewer are highly appreciated. We are hoping for pleasant response and further good comments (if any) from your side.
Dr. Abdul Qayyum
Department of Agronomy,
The University of Haripur 22620 Pakistan
******************************************************************************
We are thankful to editor and reviewers for timely completion of review process and providing us with valuable feedback.
Reviewer Comments 2
Comment: English and readability should be improved. There are expressions or words repeating twice in the same sentence. This is generally the problem occurring throughout all the manuscript. Two last sentences can be redrafted to one simple sentence.
Reply: English and readability is improved now. Corrections have been made accordingly and highlighted with blue color throughout the manuscript.
Comment: It is necessary to consult with native speaker to improve the style of the manuscript.
Reply: Consulted with native English speaker and improved the manuscript.
Comment: The typos and repeating words are quite often.
Reply: Typos and repeating words are removed from the manuscript.
24-27: There are some info about methods, and MS abbreviation is not explained even in other part of manuscript, these should be placed in another place, moreover, in abstract some particular values – most important findings - should be mentioned.
Reply: Necessary corrections have been made. Information about MS is provided now and material and methods is explained well now. Abstract is restructured and important findings are placed and highlighted with blue color. See line # 26-37.
Introduction: 41: … diminishing. Reference is missing here.
47: … environment Reference is missing here.
Reply: References have been added and highlighted with blue color. See line # 49.
55: … production to fulfill …
Reply: Correction has been done and highlighted with blue color. See line # 55-56.
Q: I am not familiar with the agricultural conditions or growing land pre-dispositions in the country of the authors, however, I would like to give opinion to foliar applications of nanoparticles. There are critical studies on the toxicity of nanoparticles, as the authors correctly reflect in Chapter 3.2. It would be worthwhile to include in the article the comments from the poly-cultural agriculture. Today, it has been found that growing one cultivar on a large area is not sustainable. The combination of approaches can then be more effective and beneficial even with a partial involvement of nanotechnology, but nanotechnology itself will hardly solve the issue without negative side effects.
Reply: Nanotechnology has the potential to address current agricultural issues in resource and environment friendly manner. Exploring new avenues and ideas for the betterment are quiet necessary to meet the current challenges. Appropriate and adequate concentration of NPs induced beneficial effects while higher doses caused toxicity. Even nitrogen is major necessary element for crop growth and development however, higher dose caused toxic effects. Now the studies have confirmed the crucial and incremental role in plant growth and development through application of NPs. Due to significant increase in higher usage of agrochemicals, environmental pollution is increasing day by day. Nanotechnology can help to increase the efficiency of used agrochemicals and reduced the quantum of agrochemicals. Therefore, one cannot ignore the potential role of nanotechnology.
Novelty: Given that the authors cite their previous work, where they also address the application of nanosilver in the same concentration on wheat (2015), it would be appropriate to emphasize what is the benefit and novelty of this current research.
Reply: According to my opinion the experimental material and results are sufficient for the justification of this study. In previous work only the effect of Ag NPs have been reported however; in this work relationship of antioxidant enzyme activity have been established in response to both Ag NPs and Au NPs.
Methods: ppm units should be converted to mg/l
Reply: Suggestions incorporated throughout the manuscript and highlighted with blue color.
English must be improved, there is “solution” repeated 3x in the sentence, etc., dilutions are mentioned in the abstract, but I have expected these info about concentrations used right here in methods.
Reply: Corrections have been made. Now the manuscript is improved. Corrections are highlighted with blue color and manuscript is read by a native speaker.
There are no info about the type of SEM used for identification of NPs and nothing about measuring conditions (besides 15 keV from the fig) and how the samples (colloids) were prepared, the images are in a bad resolution, it is not clear if the regime was transmission and dark field (however if such, 15 keV is not enough). The scale is not appropriate and it is not obvious according to micrographs that we see nanoparticles, in the case of silver there are quite big particles or aggregates… The scales for gold and silver are different. It will be helpful to do this measurements again and provide readers with relevant images with nano-metric scale or some details of shapes and size of both types of nanoparticles. Scanning electron microscopy is not appropriate method, scanning transmission EM (STEM) is much better choice with 30 keV or standard transmission electron microscopy with e.g. 100 keV. For nanoparticles characterization other methods such as UV-Vis are useful to use to control the size and stability, despite the color of the colloids seems to be ok. However the relevant EM images are crucial. In this form fig 1 and 2 cannot be presented. Moreover these info about size and related figs should be in the part Results and discussion. For gold nanoparticles preparation there is no reference cited…
Reply: It is cleared from the previous studies that nanoparticles with larger size have stress inducing and growth retardant effects while in our study both nanoparticles have higher antioxidant enzyme activity than in control. As well as size is concern same protocol was followed by other scientist in same lab (Jhanzab et al., 2019), therefore NPs were as fine particles with round shape in symmetrical arrangement as mentioned in SEM image. As well as SEM image is concern, we are not self sufficient in the facilities, we have only SEM facility that expressed in this paper, not any other alternate facility with 30 Kev.
129: Institute
Reply: It has been added and highlighted with blue color. See line # 104-105; 117-118.
136: … as described above … in the abstract? MS should be explained and the components of this medium/solution should be specified in this chapter with some reference. See comment 24-27 also.
Reply: Corrections have been made and highlighted with blue color. See line # 131.
140: inches are not SI unit 150: Why T? What does it mean? Further in the text it is not used, it is an internal designation of samples? I think it is useless, the concentration were mentioned for both metals previously.
Reply: Corrections have been made and highlighted with blue color. See line # 134.
171-177: It will be helpful to define why these enzymes are measured. Usually it is suitable to describe not only what method was chosen but also why and what this method/measurement should tell us about the sample/process.
Reply: These enzymes determine the growth and development of the plants. Higher antioxidant enzyme activity will lead to significant higher increase growth and development. Now a day’s nutritionist advised to take more food having higher antioxidant enzyme activity. Actually, methods described to quantification of the enzymes from the sample. While standard protocols and procedures were opted to analyze the samples.
184: Characterization of nanoparticles
Reply: Corrections have been made.
189-197: I don’t understand the placement of these info here. Moreover, why the synthesis of Fe NPs is mentioned? Relevant images of synthesized Ag and Au nanoparticles, together with colloids and probably UV-Vis graphs compared or just commented that they are in agreement with Kulkarni and another ref for Au will be sufficient.
Reply: Corrections have been made.
3.2. Chapter is too long, subdividing to more chapters will be helpful. Moreover, this part is not well readable, because of still repeating info about increasing/decreasing observed parameters, such as describing a course of the graphs… , some info are repeating, in general this chapter could be much shorter.
Reply: Results and discussion parts is reduced and divided into various sections as suggested.
230: Concentrations instead of levels
Reply: Corrections have been made and highlighted with blue color. See line # 202.
235: … longer duration? What does it mean? What is the difference and novelty comparing these too studies? What was the time of spraying or just the process of application or contact the nanoparticles with the plant? The process of NPs application/spraying is poorly described in methods and there is no info about duration.
Reply: Sentence corrected and modified. Corrections have been made highlighted with blue color. See line # 213-214 & 245-246.
281: control plants – please use “other plants, other samples, etc.” control plant/sample is only one.
Reply: Corrections have been made and highlighted with blue color. See line # 262-263.
Please check again or reconsider the content of intro-part and results and discussion – some parts, such as 356-361, maybe could be better place in introduction or just deleted to don’t leave the topic of results discussion.
Reply: Corrections have been made.
365: 25 or 20 ppm?
Reply: 20 ppm.
369: a new subchapter about toxicity?
Reply: It has been corrected. See line # 426-449.
378 and further: this part is very poorly written, sentences are not clear and understandable.
Reply: It has been corrected. Now well written and sentences are clear.
437 vs. 442 – Guaiacol – guaiacol
Reply: Corrections have been made. See line # 413 and 421.
445-447: see 171-177
Reply: Corrections have been made.
Conclusions: What are the future plans? The testing in filed conditions?
Reply: In future the study must test under field conditions.
After some graphic improvements and additions the results/graphs, Figure 3 could be a graphic abstract of the article, otherwise I do not consider it is important in the text in this state.
Reply: Fig. 3 has been removed.
Tables: I'm wondering if it wouldn't be worth converting some of the data from the tables into graphical dependencies.
Reply: Right now I have put all the data in the form of tables not in figures. Most of the time both forms are acceptable. So I want to keep the data in tabulated form. If it’s too necessary then it can be converted to figures form.
Round 2
Reviewer 2 Report
I thank the authors for editing the manuscript according to the opponents' recommendations, in the current form the quality of the article has been moved to a higher level. Regarding English, I just want to point out that I am not a native speaker, so I cannot objectively assess the improvement, but as the authors claim, the manuscript has already undergone a linguistic correction. Only on Line 287 I noticed "Chlorophyll" with a "Ch".
Regarding the answers to the questions, I would like to point out that I appreciate the scientific discussion that was supposed to be stimulated by these questions, and I would appreciate if the authors incorporate their opinions and answers in a certain form into the text next time and not address it only to the opponent. This concerns the answer to the importance of nanotechnology in agriculture, the novelty of research and the answer to 171-177 and future plans… Regarding the importance of the study of enzymes (171-177) I just want to add that explaining the methodology is important for readers from other disciplines.
I agree that the success and yield of plant cultivation is always associated with the concentration of whether agrochemicals, nanoparticles or trace elements. It is necessary to look for the lowest but still effective concentrations for the given case and to avoid large-scale applications. The importance of biodiversity, poly-cultures and crop rotation cannot be overlooked. Finding new ways is as important as returning to traditional practices.
As for the SEM images, could the name / type of the device (manufacturer) be added at least? Unfortunately, it is not possible to determine the size of nanoparticles from the submitted images, moreover, two size fractions are visible for silver, if they are not aggregates, resolution (magnification) is insufficient for correct sizing, it is possible that values ​​are taken from already cited publications, then citations should be included again. In my opinion, the shape, size (as noted by the authors) and the stability of nanoparticles and basically the method of their preparation (stabilizing components) affect the bioavailability of particles, so I emphasize a more thorough study of the nature of nanoparticles at least by imaging method, better supplemented UV-Vis next time.
Author Response
Dated: 22nd July 2021
Dear Editor,
Greetings,
Thank you very much for your time and comments regarding our manuscript (agronomy-1272414). Our manuscript “Relationship of Antioxidant Enzyme Activity with Growth Parameters in Response to Silver and Gold Nanoparticles in Wheat (Triticum aestivum L.)” has been revised carefully and here we are giving our response to the reviewers’ comments. We have improved the manuscript according to the reviewers’ comments and suggestions. All the revisions can be easily identified from manuscript highlighted with RED color.
Once again thanks for your co-operation and valuable comments and suggestion. Moreover, the efforts of the reviewer are highly appreciated. We are hoping for pleasant response and further good comments (if any) from your side.
Dr. Abdul Qayyum
Department of Agronomy,
The University of Haripur 22620 Pakistan
******************************************************************************
We are thankful to editor and reviewers for timely completion of review process and providing us with valuable feedback.
Reviewer Comments 2
Comment: I thank the authors for editing the manuscript according to the opponents' recommendations, in the current form the quality of the article has been moved to a higher level. Regarding English, I just want to point out that I am not a native speaker, so I cannot objectively assess the improvement, but as the authors claim, the manuscript has already undergone a linguistic correction. Only on Line 287 I noticed "Chlorophyll" with a "Ch".
Reply: Thanks for appreciation. I have incorporated all the comments as suggested. If you suggest any more, I am happily to incorporate those too.
Comment: Regarding the answers to the questions, I would like to point out that I appreciate the scientific discussion that was supposed to be stimulated by these questions, and I would appreciate if the authors incorporate their opinions and answers in a certain form into the text next time and not address it only to the opponent. This concerns the answer to the importance of nanotechnology in agriculture, the novelty of research and the answer to 171-177 and future plans… Regarding the importance of the study of enzymes (171-177) I just want to add that explaining the methodology is important for readers from other disciplines.
Reply: Enzymes methodology is briefly explained from line no. 169-175. Highlighted with yellow color. These are the standard protocols that’s why I have discussed in detail. That’s the main reason.
Comment: I agree that the success and yield of plant cultivation is always associated with the concentration of whether agrochemicals, nanoparticles or trace elements. It is necessary to look for the lowest but still effective concentrations for the given case and to avoid large-scale applications. The importance of biodiversity, poly-cultures and crop rotation cannot be overlooked. Finding new ways is as important as returning to traditional practices.
Reply: yes off course the importance of biodiversity, poly-cultures and crop rotation cannot be overlooked. We have not worked on these aspects. In future these aspects will be kept in mind before conducting a study. In current project, our only focus was to analyze the protective effect of Ag and Au NPs against oxidative stress in relation to biomass, leaf area and chlorophyll content of wheat seedlings.
Comment: As for the SEM images, could the name / type of the device (manufacturer) be added at least? Unfortunately, it is not possible to determine the size of nanoparticles from the submitted images, moreover, two size fractions are visible for silver, if they are not aggregates, resolution (magnification) is insufficient for correct sizing, it is possible that values ​​are taken from already cited publications, then citations should be included again. In my opinion, the shape, size (as noted by the authors) and the stability of nanoparticles and basically the method of their preparation (stabilizing components) affect the bioavailability of particles, so I emphasize a more thorough study of the nature of nanoparticles at least by imaging method, better supplemented UV-Vis next time.
Reply: Type of the device (manufacturer) is added. See line no 106-107 & 123. Highligthed with yellow color.